# Approximate Triangulations of Grassmann Manifolds

**Kevin P. Knudson** 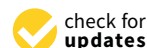

Department of Mathematics, University of Florida, Gainesville, FL 32611, USA; kknudson@ufl.edu

**Abstract:** We define the notion of an approximate triangulation for a manifold $M$ embedded in Euclidean space. The basic idea is to build a nested family of simplicial complexes whose vertices lie in $M$ and use persistent homology to find a complex in the family whose homology agrees with that of $M$. Our key examples are various Grassmann manifolds $G_k(\mathbb{R}^n)$.

**Keywords:** Grassmannian; persistent homology; Vietoris–Rips complex; witness complex; triangulation

## 1. Introduction

Smooth manifolds admit piecewise-linear triangulations [1]. However, there are many subsequent questions one might ask: How many simplices are required? What is the minimal number of vertices? Is there an algorithm to construct a triangulation?

A great deal of work in algebraic topology has been devoted to these topics. The question of the number of simplices required to triangulate a given manifold is often attacked by sophisticated cohomological methods involving characteristic classes (such arguments also often yield estimates on the minimal embedding dimension for the manifold). Surprisingly, much of this work is very recent [2,3]. A main result in [3] is the following.

**Theorem 1** ([3], Theorem 3.10). *Every triangulation of the Grassmann manifold $G_k(\mathbb{R}^{n+k})$ must have at least*

$$[(n + k)(n + k + 1) - 2kn] \cdot (2^{kn+1} - 1)$$

*simplices.*

For example, any triangulation of the manifold $G_2(\mathbb{R}^4)$ must have at least 372 simplices. The Grassmann manifolds will be defined in Section 2.1 below. These are important spaces to study because of their utility in algebraic topology, especially with respect to the study of characteristic classes [4]. Moreover, while the theory of these spaces is rich, there are still many unsolved questions about them. For example, if we pass to the limit as $n \to \infty$, we obtain an infinite Grassmannian $G_k(\mathbb{R}^\infty)$; it is still unknown [5] if these spaces are triangulable for $k \geq 3$. If one could construct compatible triangulations of the various $G_k(\mathbb{R}^n)$, then one might be able to answer this question. Our methods here do not apply, however.

Unfortunately, most results along these lines are *not* constructive; that is, the proofs do not yield an explicit triangulation of the manifold. In fact, if one seeks a triangulation of a Grassmannian $G_k(\mathbb{R}^n)$, the end result is usually disappointment. For the smallest nontrivial space, $G_1(\mathbb{R}^3) = \mathbb{R}P^2$, there are many well-known small triangulations, and even an algorithm to generate a triangulation from any collection of points in general position [6]. Beyond that, however, results are sparse.

In this paper, we develop a procedure to find what we call an *approximate triangulation* of the manifold $G_k(\mathbb{R}^n)$ (Definition 3). The basic idea is to first generate a sample of points on $G_k(\mathbb{R}^n)$.

This already leads to technical difficulties involving embeddings of these spaces into a Euclidean space $\mathbb{R}^N$, but we are able to solve this. We then build a nested family of simplicial complexes on the point cloud, parametrized by the positive real numbers. The persistent homology of this family is then computed and we identify an interval of parameters for which the mod 2 homology of the complexes in that range agrees with that of $G_k(\mathbb{R}^n)$. Such a complex is then a viable model for the manifold: its vertices lie in $G_k(\mathbb{R}^n) \subset \mathbb{R}^N$ and it has the correct mod 2 homology. (As a further check, we can compute the mod 3 homology to ensure it is correct as well if we wish.) We then implement this procedure for the following spaces: $\mathbb{R}P^2 \subset \mathbb{R}^4$, $\mathbb{R}P^2 \subset \mathbb{R}^5$, $\mathbb{R}P^3 \subset \mathbb{R}^9$, and $G_2(\mathbb{R}^4) \subset \mathbb{R}^{16}$. Computational limitations have so far prohibited further calculations; we discuss this in Section 4.

In effect, what we are doing is taking a tool developed for examining unknown objects (persistent homology modules built from point clouds) and using it on spaces that are well-understood (Grassmann manifolds) to build new structures associated with those spaces (approximate triangulations). We have chosen the Grassmannians because their homology is relatively easy to compute and they arise in numerous areas in mathematics (topology, algebraic geometry, combinatorics). These same ideas could be applied to any manifold whose homology groups are known a priori to produce an approximate triangulation, which then might be useful for other applications (e.g., storing the manifold efficiently in a computer).

## 2. Materials and Methods

Further details and proofs of the results in Sections 2.1 and 2.2 may be found in [4].

### 2.1. Grassmann Manifolds

Denote by $\mathbb{R}^n$ the Euclidean space of dimension $n$. By a *k-frame* in $\mathbb{R}^n$, we mean a $k$-tuple of linearly independent vectors; denote by $V_k(\mathbb{R}^n)$ the collection of $k$-frames in $\mathbb{R}^n$. This is an open subset of the $k$-fold Cartesian product $\mathbb{R}^n \times \cdots \times \mathbb{R}^n$.

**Definition 1.** *The* Grassmann manifold $G_k(\mathbb{R}^n)$ *is the set of all k-dimensional planes through the origin in $\mathbb{R}^n$. It is topologized via the quotient map $V_k(\mathbb{R}^n) \to G_k(\mathbb{R}^n)$ which takes a k-frame to the k-plane it spans.*

When $k = 1$, we see that $G_1(\mathbb{R}^n)$ is the real projective space $\mathbb{R}P^{n-1}$, a manifold of dimension $n - 1$. In general, we have the following result.

**Lemma 1.** *The Grassmannian $G_k(\mathbb{R}^n)$ is a compact manifold of dimension $k(n - k)$. The map $X \to X^\perp$, which takes a k-plane to its orthogonal complement is a diffeomorphism between $G_k(\mathbb{R}^n)$ and $G_{n-k}(\mathbb{R}^n)$.*

### 2.2. Schubert Cells

Grassmannians have a well-known cell decompostion into Schubert cells. Consider the sequence of subspaces of $\mathbb{R}^n$: $\mathbb{R}^0 \subset \mathbb{R}^1 \subset \mathbb{R}^2 \subset \cdots \subset \mathbb{R}^n$, where $\mathbb{R}^i$ consists of the vectors of the form $(a_1, \ldots, a_i, 0, \ldots, 0)$. Any $k$-plane $X$ gives rise to a sequence of integers

$$0 \le \dim(X \cap \mathbb{R}^1) \le \dim(X \cap \mathbb{R}^2) \le \cdots \le \dim(X \cap \mathbb{R}^n) = k.$$

Consecutive integers differ by at most 1.

**Definition 2.** *A Schubert symbol $\sigma = (\sigma_1, \ldots, \sigma_k)$ is a sequence of k integers satisfying*

$$1 \le \sigma_1 < \sigma_2 < \cdots < \sigma_k \le n.$$

Given a Schubert symbol $\sigma$, let $e(\sigma) \subset G_k(\mathbb{R}^n)$ denote the set of $k$-planes $X$ such that

$$\dim(X \cap \mathbb{R}^{\sigma_i}) = i, \dim(X \cap \mathbb{R}^{\sigma_i - 1}) = i - 1.$$

Each $X \in G_k(\mathbb{R}^n)$ belongs to precisely one of the sets $e(\sigma)$.

**Lemma 2.** *$e(\sigma)$ is an open cell of dimension $d(\sigma) = (\sigma_1 - 1) + (\sigma_2 - 2) + \cdots + (\sigma_k - k)$.*

In terms of matrices, $X \in e(\sigma)$ if and only if it can be described as the row space of a $k \times n$ matrix of the form

$$
\begin{bmatrix}
* & \cdots & * & 1 & 0 & \cdots & 0 & 0 & 0 & \cdots & 0 & 0 & 0 & \cdots & 0 \\
* & \cdots & * & * & * & \cdots & * & 1 & 0 & \cdots & 0 & 0 & 0 & \cdots & 0 \\
\vdots & & & & & & & & & & & & & & \vdots \\
* & \cdots & * & * & * & \cdots & * & * & * & \cdots & * & 1 & 0 & \cdots & 0
\end{bmatrix}
$$

where the $i$-th row has $\sigma_i$-th entry positive (say equal to 1) and all subsequent entries zero. Equivalently, we could (and do in the sequel) consider the column space of the transpose of this matrix.

For example, the possible Schubert symbols and cells for $G_2(\mathbb{R}^4)$ are as follows. Such a symbol has the form $\sigma = (\sigma_1, \sigma_2)$ where $1 \leq \sigma_1 < \sigma_2 \leq 4$.

| $\sigma$ | $d(\sigma)$ |
|---|---|
| $(1,2)$ | 0 |
| $(1,3)$ | 1 |
| $(1,4)$ | 2 |
| $(2,3)$ | 2 |
| $(2,4)$ | 3 |
| $(3,4)$ | 4 |

**Theorem 2.** *The $\binom{n}{k}$ sets $e(\sigma)$ form the cells of a CW-decomposition of $G_k(\mathbb{R}^n)$.*

**Proposition 1.** *The number of $r$-cells in $G_k(\mathbb{R}^n)$ is equal to the number of partitions of $r$ into at most $k$ integers, each of which is $\leq n - k$.*

The mod 2 homology of $G_k(\mathbb{R}^n)$ is easily computed from the Schubert cell decomposition: since the induced boundary maps are all either 0 or multiplication by 2, the mod 2 homology has a basis corresponding to the cells.

Continuing the example of $G_2(\mathbb{R}^4)$, we have

$$
H_i(G_2(\mathbb{R}^4), \mathbb{Z}/2) = \begin{cases}
\mathbb{Z}/2 & i = 0 \\
\mathbb{Z}/2 & i = 1 \\
\mathbb{Z}/2 \oplus \mathbb{Z}/2 & i = 2 \\
\mathbb{Z}/2 & i = 3 \\
\mathbb{Z}/2 & i = 4
\end{cases}
$$

Since we will need it below, we also note the integral homology of $G_2(\mathbb{R}^4)$ [7]:

$$
H_i(G_2(\mathbb{R}^4), \mathbb{Z}) = \begin{cases}
\mathbb{Z} & i = 0 \\
\mathbb{Z}/2 & i = 1, 2 \\
0 & i = 3 \\
\mathbb{Z} & i = 4
\end{cases}
$$

Using the Universal Coefficient Theorem, one then quickly deduces that the homology groups $H_i(G_2(\mathbb{R}^4), \mathbb{Z}/3)$ are $\mathbb{Z}/3$ for $i = 0, 4$ and 0, otherwise.

### 2.3. Persistent Homology

Suppose we are given a finite nested sequence of finite simplicial complexes

$$K_{R_1} \subset K_{R_2} \subset \cdots \subset K_{R_p},$$

where the $R_i$ are real numbers $R_1 < R_2 < \cdots < R_p$. For each homological degree $\ell \geq 0$, we then obtain a sequence of homology groups and induced linear transformations (homology with $\mathbb{Z}/2$-coefficients for simplicity)

$$H_\ell(K_{R_1}) \to H_\ell(K_{R_2}) \to \cdots \to H_\ell(K_{R_p}).$$

Since the complexes are finite, each $H_\ell(K_{R_i})$ is a finite-dimensional vector space. Thus, there are only finitely many distinct homology classes. A particular class $z$ may come into existence in $H_\ell(K_{R_s})$, and then one of two things happens. Either $z$ maps to 0 (i.e., the cycle representing $z$ gets filled in) in some $H_\ell(K_{R_t})$, $R_s < R_t$, or $z$ maps to a nontrivial element in $H_\ell(K_{R_p})$. This yields a *barcode*, a collection of interval graphs lying above an axis parametrized by $R$. An interval of the form $[R_s, R_t]$ corresponds to a class that appears at $R_s$ and dies at $R_t$. Classes that live to $K_{R_p}$ are usually represented by the infinite interval $[R_s, \infty)$ to indicate that such classes are real features of the full complex $K_{R_p}$.

As an example, consider the boundary of the tetrahedron $T$ with filtration

$$T_0 \subset T_1 \subset T_2 \subset T_3 \subset T_4 \subset T_5 = T$$

defined by $T_0 = \{v_0, v_1, v_2, v_3\}$, $T_1 = T_0 \cup \{\text{all edges}\}$, $T_2 = T_1 \cup [v_0 v_1 v_2]$, $T_3 = T_2 \cup [v_0 v_1 v_3]$, $T_4 = T_3 \cup [v_0 v_2 v_3]$, and $T_5 = T$ (this is topologically a 2-sphere). The barcodes for this filtration are shown in Figure 1. Note that, initially, there are four components ($\beta_0 = 4$), which get connected in $T_1$, when three independent 1-cycles are born ($\beta_1 = 3$). These three 1-cycles die successively as triangles get added in $T_2$, $T_3$, and $T_4$. The addition of the final triangle in $T_5$ creates a 2-cycle ($\beta_2 = 1$).

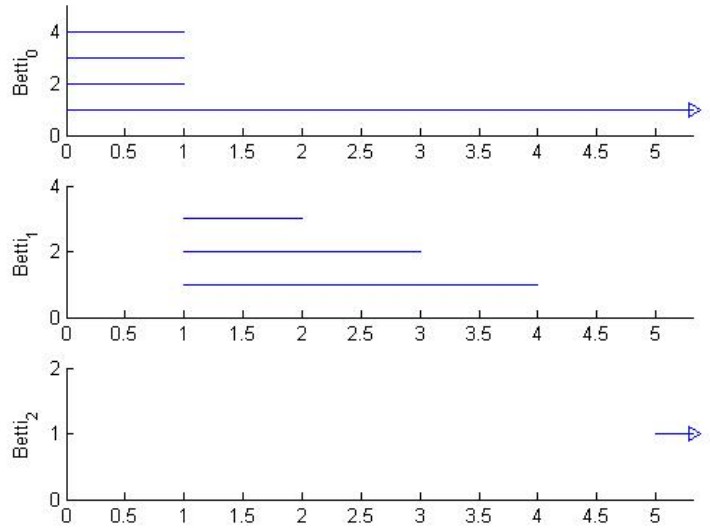

**Figure 1.** The barcodes for a filtration of the boundary of the tetrahedron.

For analyzing point cloud data, one needs a simplicial complex modeling the underlying space. Since it is impossible to know a priori if a complex is "correct", one builds a nested family of complexes approximating the data cloud, computes the persistent homology of the resulting filtration, and looks for homology classes that exist in long sections of the filtration. We discuss two popular methods for doing this in the next subsection.

### 2.4. Vietoris–Rips and Witness Complexes

Now suppose we are given a discrete set $X$ of points in some metric space (typically a Euclidean space $\mathbb{R}^m$). The standard example of such an object is a sample of points from some geometric object $M$. We would like to recover information about $M$ from the sample $X$, and the first step is to obtain an approximation of $M$ using only the point cloud $X$. There are many such techniques; perhaps the most classical is the *Delaunay triangulation* of $X$. This is defined as follows. Say $X = \{x_1, x_2, \ldots, x_r\} \subset \mathbb{R}^m$. The *Voronoi decomposition* of $\mathbb{R}^m$ relative to $X$ is the partition of $\mathbb{R}^m$ into cells $V(x_i)$, $i = 1, \ldots, r$, defined by

$$V(x_i) = \{x \in \mathbb{R}^m : ||x - x_i|| \leq ||x - x_j||, j \neq i\}.$$

The corresponding Delaunay triangulation, $\text{Del}(X)$, is the nerve of the Voronoi decomposition; that is, a collection $V(x_{i_0}), \ldots, V(x_{i_\ell})$ forms an $\ell$-simplex in $\text{Del}(X)$ if $\cap_{j=0}^{\ell} V(x_{i_j}) \neq \varnothing$. One obtains a geometric realization of $\text{Del}(X)$ via the map $V(x_i) \mapsto x_i$. See Figure 2 for an example.

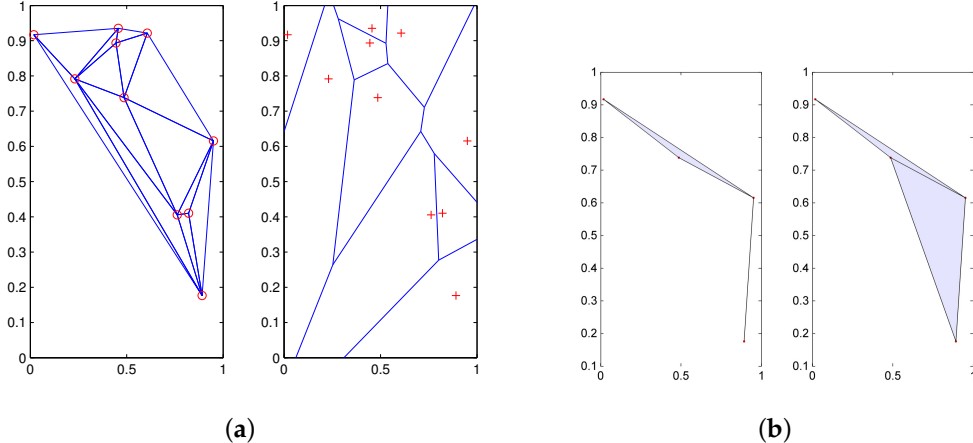

(a)                                  (b)

**Figure 2.** (**a**) a Delaunay triangulation of a collection of points in the plane with the corresponding Voronoi diagram, and (**b**) two associated witness complexes.

While the Delaunay triangulation provides a good approximation to the underlying space $M$, it has several disadvantages. If the point cloud $X$ is large, there will be a very large number of simplices in $\text{Del}(X)$. In addition, $\text{Del}(X)$ suffers from the "curse of dimensionality;" that is, if the ambient dimension ($m$) is large, calculating the Voronoi decomposition is computationally expensive.

There are many popular alternatives to the Delaunay triangulation. The one used most often is the *Vietoris–Rips complex*, which is built as follows. Consider the point cloud $X$ and let $r > 0$. The Vietoris–Rips complex with parameter $r$ is the simplicial complex $VR(X, r)$ whose $k$-simplices are

$$\{(x_0, \ldots, x_k) : d(x_i, x_j) < r, i \neq j\}.$$

That is, if one imagines a ball of radius $r/2$ around each point $x \in X$, then we join the points $x_i$ and $x_j$ with an edge if the balls intersect. Observe that if $r < r'$ then there is an inclusion of complexes $VR(X, r) \subset VR(X, r')$. We therefore have a nested sequence of complexes $\{VR(X, r)\}_{r \geq 0}$ and we may study the persistent homology of this filtration. The corresponding barcodes yield information about the topology of the underlying space $M$.

Many software packages support the calculation of Vietoris–Rips persistence on point clouds. In this paper, we use the Eirene package developed by Gregory Henselman [8]. Other popular programs include Ulrich Bauer's Ripser [9] and Vidit Nanda's Perseus [10].

In Section 3.5, we shall use the *witness complexes* of de Silva and Carlsson [11]. The idea is to model the Delaunay triangulation on a smaller set of points $L \subset X$, called *landmarks*, in such a way that the topology of the underlying object is well-approximated. Moreover, the definition makes sense

in any metric space, so assume that $X$ is a metric space with distance function $d$ (e.g., $X$ could be a finite point cloud in $\mathbb{R}^m$ with the usual Euclidean distance). Choose a subset $L = \{\ell_1, \ell_2, \ldots, \ell_n\}$ of $X = \{x_1, x_2, \ldots, x_N\}$ and let $R \geq 0$ be a real number.

The *witness complex* $W(X, L, R)$ is defined as follows:

- The vertex set of $W(X, L, R)$ is $L$;
- $\ell, \ell' \in L$ span an edge if there exists an $x \in X$, called a *witness*, such that

$$d(x, \ell), d(x, \ell') \leq R + \min\{d(x, \ell'') : \ell'' \in L - \{\ell, \ell'\}\};$$

- A collection $\ell_0, \ldots, \ell_p \in L$ spans a $p$-simplex if $\{\ell_i, \ell_j\}$ span an edge for all $i \neq j$.

Examples of witness complexes are shown in Figure 2b alongside the associated Delaunay triangulation. Four landmark points were chosen using the maxmin procedure described below. The complex on the left has $R = 0.0329$, and the complex on the right has $R = 0.1317$. Note that the larger value of $R$ yields a complex with more simplices. In addition, note that the witness complex is a coarse approximation of the Delaunay triangulation.

We make some observations about this definition. Let $D$ be the $n \times N$ matrix of distances from points in $L$ to points in $X$:

- If $R = 0$, then $\ell, \ell' \in L$ form an edge if there is an $x_i \in X$ such that $d(x_i, \ell)$ and $d(x_i, \ell')$ are the two smallest entries in the $i$-th column of $D$. This is analogous to the existence of an edge in the Delaunay triangulation Del($L$).
- For $R > 0$, one may think of relaxing the boundaries of the Voronoi diagram of $L$ and taking the nerve of the resulting covering of $X$.
- If $0 \leq R < R'$, then there is an inclusion of simplicial complexes $W(X, L, R) \subseteq W(X, L, R')$.

By a theorem of de Silva and Carlsson [11], this complex is a natural analogue of the Delaunay triangulation for a space represented by point cloud data.

Suppose that $X$ is a sample of points from some object $M \subset \mathbb{R}^m$. There is no guarantee that $W(X, L, R)$ recovers the topology of $M$, but experiments on familiar geometric objects [11] (spheres, for example) suggest that, for a suitable range of values of $R$ and good choices of landmarks $L$, the topology of $W(X, L, R)$ is the same as that of $M$. This begs the question:

1. How should the landmark set $L$ be chosen?
2. What is the correct value of $R$?

The second question is best handled via the use of persistent homology, which we discussed in Section 2.3 above. As for the choice of landmarks, there are three standard options:

1. Select landmarks at random.
2. Use the *maxmin* procedure: Choose a seed $\ell_1$ at random. Then, if $\ell_1, \ldots, \ell_n$ have been chosen, let $\ell_{n+1} \in X - \{\ell_1, \ldots, \ell_n\}$ be the point which maximizes the function

$$z \mapsto \min\{d(z, \ell_1), d(z, \ell_2), \ldots, d(z, \ell_n)\}.$$

3. Use a density-based strategy.

The maxmin procedure yields more evenly-spaced landmarks, but tends to emphasize extremal points. It is generally more reliable than a random selection [11]. Another useful resource is [12]. In our experiments in Section 3.5 below, we use the maxmin process to generate landmarks.

*2.5. Sampling Procedures*

To build a Vietoris–Rips or witness complex on points in $G_k(\mathbb{R}^n)$, we need to develop a sampling procedure. The first question to be asked is in which Euclidean space do we embed $G_k(\mathbb{R}^n)$? This is highly nontrivial. Even in the case of projective spaces ($k = 1$), it is not so obvious how to proceed. A whole industry has been devoted to the question of the minimal embedding dimension of $\mathbb{R}P^n$ [13], but the proof of the minimality of any particular embedding rarely comes with an explicit *formula* for the map. An exception is if one insists on an *isometric* embedding [14], but the minimal dimension of such an embedding for $\mathbb{R}P^n$ is $n(n+3)/2$, which grows rather quickly.

For arbitrary Grassmannians, one could try to use the Plücker embedding $G_k(\mathbb{R}^n) \to P(\bigwedge^k(\mathbb{R}^n)) = \mathbb{R}P^{\binom{n}{k}-1}$ defined by

$$(x_1, \ldots, x_k) \mapsto [x_1 \wedge \cdots \wedge x_k]$$

(where $[v]$ denotes the line spanned by the vector $v$) and then embed the target projective space into Euclidean space. Of course, this explodes the dimension further, making this an impractical solution. Aside from some low dimensional projective spaces, we will instead approach this problem via the following result.

**Proposition 2.** *The manifold $G_k(\mathbb{R}^n)$ is diffeomorphic to the smooth manifold consisting of all $n \times n$ symmetric, idempotent matrices of trace k. The map $\varphi$ realizing this takes a k-plane X to the operator defined by orthogonal projection onto X.*

**Proof.** If $X$ is a $k$-plane with orthonormal basis $x_1, \ldots, x_k$, denote by $A$ the $n \times k$ matrix having the $x_i$ as columns. Define a map $\varphi : G_k(\mathbb{R}^n) \to M_n(\mathbb{R})$ by $X \mapsto AA^T$. This map is clearly smooth since it consists of polynomials in the entries of the various $x_i$. Moreover, it is well-defined since, if $y_1, \ldots, y_k$ is another orthonormal basis of $X$ with associated matrix $B$, then there is an orthogonal matrix $O$ such that $B = AO$. Then, $BB^T = (AO)(AO)^T = AOO^T A^T = AA^T$. The matrix $AA^T$ is symmetric: $(AA^T)^T = (A^T)^T A^T = AA^T$. It is idempotent: $(AA^T)^2 = AA^T AA^T = AI_k A^T = AA^T$ (note that $A^T A = I_k$, the $k \times k$ identity matrix, since the columns of $A$ are orthonormal). Finally, the trace of $AA^T$ is $k$ since its rank is $k$ and its only eigenvalues are 0 and 1. Thus, the image of $\varphi$ lies in the set of symmetric, idempotent matrices of trace $k$. To see that $\varphi$ surjects onto this set, note that such a matrix $B$ is projection onto a $k$-dimensional subspace $X$ and there exists a basis $x_1, \ldots, x_k$ with $\varphi(X) = B$. Injectivity of $\varphi$ follows since the subspace determined by a projection is unique. □

Now, to generate a sample of points on which to build a Vietoris–Rips or witness complex, we will use the embedding $\varphi$. A crude sampling is then obtained by the following procedure:

- Select $k$ random vectors in $\mathbb{R}^n$.
- Perform the Gram–Schmidt orthogonalization algorithm to yield an orthornomal set $x_1, \ldots, x_k$. Let $A$ be the matrix with $x_i$ as columns.
- Compute $AA^T$.

One immediate problem with this process is that the $k$-plane it constructs lives in the top-dimensional Schubert cell with probability 1. However, since we know the space we are interested in, and we know its homology, we can bias our sample to ensure we include points from each Schubert cell. The following procedure implements this idea:

- Determine the percentage of sample points desired from each Schubert cell. For example, one might choose 5% from a 1-cell, 10% from a 2-cell, and so on.
- Elements of a given Schubert cell correspond to the column space of a particular matrix form. Generate such a matrix $B$ using random vectors of the required form.
- Generate a random $n \times n$ orthogonal matrix $X$.

- Add the matrix $A = X(BB^T)X^T$ to the point cloud.

Note the final step above. If we merely took the matrix $B$, we would not end up with a well-distributed sample. For example, in the case of $G_2(\mathbb{R}^4)$, such a matrix lying in the 1-cell of the Schubert decomposition has the following form:

$$B = \begin{bmatrix} 1 & * \\ 0 & 1 \\ 0 & 0 \\ 0 & 0 \end{bmatrix}$$

The corresponding point in $\mathbb{R}^{16}$ would have most coordinates equal to 0, which is clearly not what we want. Conjugating the various $BB^T$ by a random orthogonal matrix $X$ (a different $X$ for each $B$) yields a wider distribution of points in $G_k(\mathbb{R}^n)$.

The MATLAB files we used to generate samples in various projective spaces and Grassmannians are available at https://github.com/niveknosdunk/grassmann.

*2.6. Approximate Triangulations*

We are now ready to search for simplicial complexes modeling the spaces $G_k(\mathbb{R}^n)$. The procedure we employ is as follows:

- Construct a sample of points on $G_k(\mathbb{R}^n)$.
- Construct a collection of Vietoris–Rips or witness complexes on the point cloud.
- Compute the persistent homology of this filtration.
- Determine a range of parameters where the homology of the complexes agrees with that of $G_k(\mathbb{R}^n)$.

**Definition 3.** *Let $K_r$ denote either $VR(X, r)$ or $W(X, L, r)$. If there exists a parameter $r > 0$ for which the $\mathbb{Z}/2$ homology of $K_r$ agrees with that of $G_k(\mathbb{R}^n)$, then we call $K_r$ an* approximate triangulation *of $G_k(\mathbb{R}^n)$.*

Note that $K_r$ is a subcomplex of the Euclidean space in which we have embedded $G_k(\mathbb{R}^n)$. However, it does *not* necessarily lie inside the embedded $G_k(\mathbb{R}^n)$. Still, its vertices do lie on $G_k(\mathbb{R}^n)$ and so we can think of this as being close to a triangulation of this manifold. For further verification, we can also compute the $\mathbb{Z}/3$ homology of $K_r$ and check it against that of $G_k(\mathbb{R}^n)$.

## 3. Results

*3.1. $\mathbb{R}P^2$, Part I*

Let us begin by embedding $\mathbb{R}P^2$ into $\mathbb{R}^4$ using the map $\psi : S^2 \to \mathbb{R}^4$ defined by

$$\psi : (x, y, z) \mapsto (xy, xz, y^2 - z^2, 2yz).$$

Note that $\psi(-x, -y, -z) = \psi(x, y, z)$ and so it descends to a map $\mathbb{R}P^2 \to \mathbb{R}^4$. Generate a sample of 100 points on $S^2$ and then use this map to get the points in $\mathbb{R}^4$. The persistence diagrams are shown in Figure 3. There is a tiny window, around $r = 0.87$, where we get the correct homology.

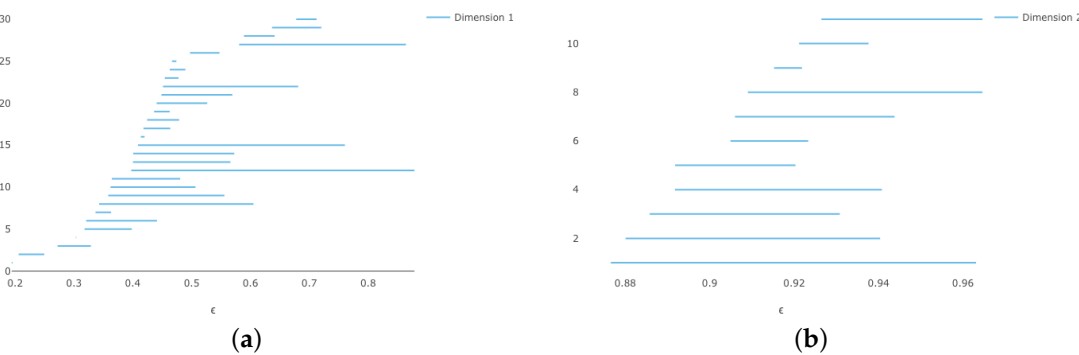

**Figure 3.** Vietoris–Rips persistence diagrams for 100 points on $\mathbb{R}P^2$ (**a**) $H_1$ persistence and (**b**) $H_2$ persistence.

Now generate a sample of 200 points. As expected, the Vietoris–Rips complex has the correct homology for a longer range of parameters, as indicated in Figure 4. Here, we see a long interval $0.69 < r < 0.87$ where we get the correct homology. Thus, the Vietoris–Rips complex built on these 200 points in $\mathbb{R}^4$ is a good approximation to $\mathbb{R}P^2$.

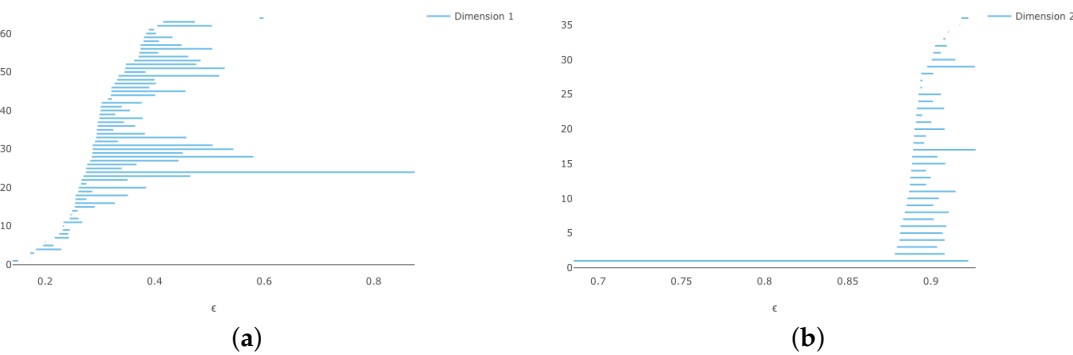

**Figure 4.** Vietoris–Rips persistence diagrams for 200 points on $\mathbb{R}P^2$ (**a**) $H_1$ persistence and (**b**) $H_2$ persistence.

### 3.2. $\mathbb{R}P^2$, Part II

The embedding of $\mathbb{R}P^2$ into $\mathbb{R}^4$ is not an isometric embedding, though. For that, we need $\mathbb{R}^5$:

$$(x, y, z) \mapsto \left( yz, xz, xy, \frac{1}{2}(x^2 - y^2), \frac{1}{2\sqrt{3}}(x^2 + y^2 - 2z^2) \right)$$

If we then generate 100 random points on this surface, we obtain the Vietoris–Rips barcodes in Figure 5. This works better than the embedding into $\mathbb{R}^4$; we get the correct answer for $0.625 < r < 0.871$. The result for 200 points is even better, and is shown in Figure 6.

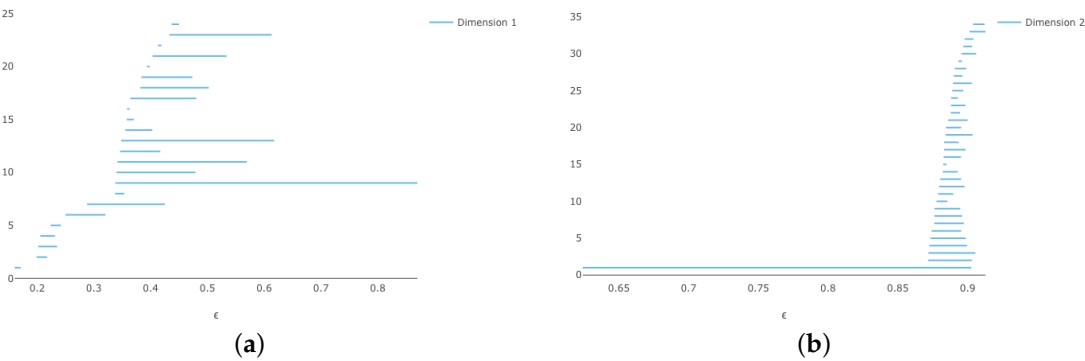

(a)

(b)

**Figure 5.** Vietoris–Rips persistence diagrams for 100 points on $\mathbb{R}P^2$, using the isometric embedding into $\mathbb{R}^5$ (**a**) $H_1$ persistence and (**b**) $H_2$ persistence.

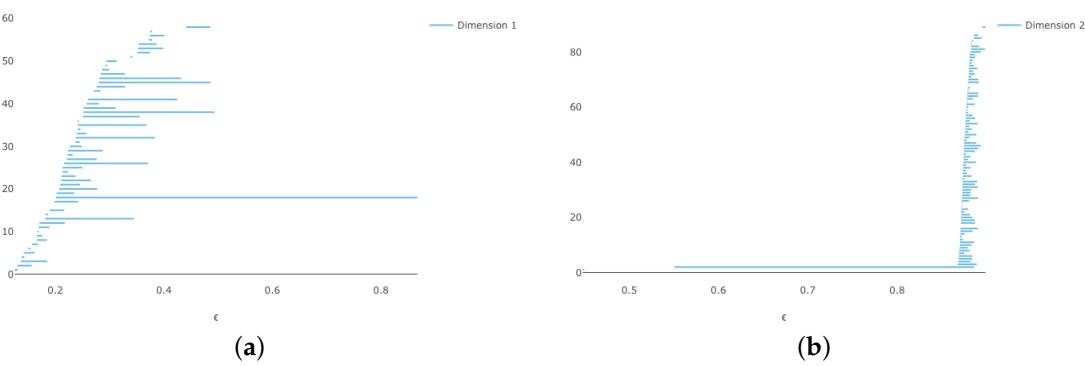

(a)

(b)

**Figure 6.** Vietoris–Rips persistence diagrams for 200 points on $\mathbb{R}P^2$, using the isometric embedding into $\mathbb{R}^5$ (**a**) $H_1$ persistence and (**b**) $H_2$ persistence.

### 3.3. $\mathbb{R}P^3$

We use the fact that $\mathbb{R}P^3$ is diffeomorphic to $SO(3)$, the space of $3 \times 3$ orthogonal matrices of determinant 1. If we select 100 random points on this space in $\mathbb{R}^9$, we find that there is only a tiny window where $\beta_2 = 1$, so 100 points probably is not enough to yield a good approximate triangulation. The barcodes are shown in Figures 7 and 8.

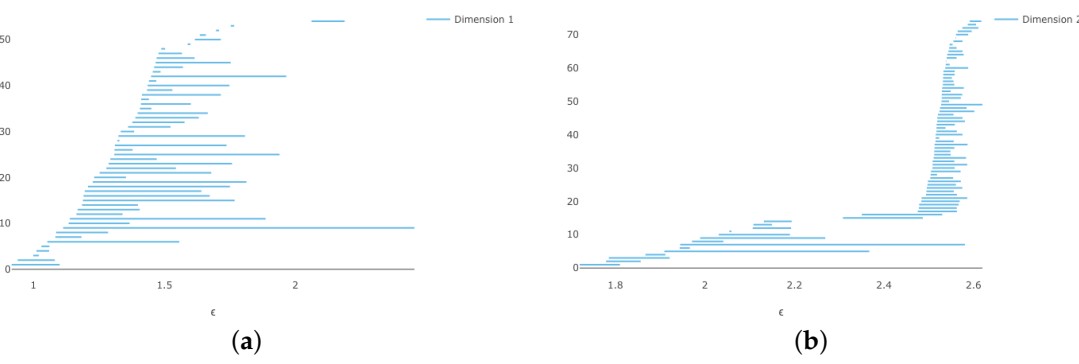

(a)

(b)

**Figure 7.** Vietoris–Rips persistence diagrams for 100 points on $\mathbb{R}P^3$, realizing it as the Lie group $SO(3) \subset \mathbb{R}^9$ (**a**) $H_1$ persistence and (**b**) $H_2$ persistence.

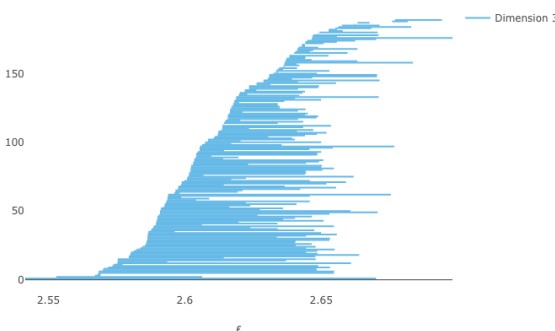

**Figure 8.** The $H_3$ barcode for 100 points on $\mathbb{R}P^3$.

If we now sample 200 points at random on $\mathbb{R}P^3$ (computation time 6:54), we obtain the barcodes in Figures 9 and 10. Note that we get the correct homology for $2.1 < r < 2.4$.

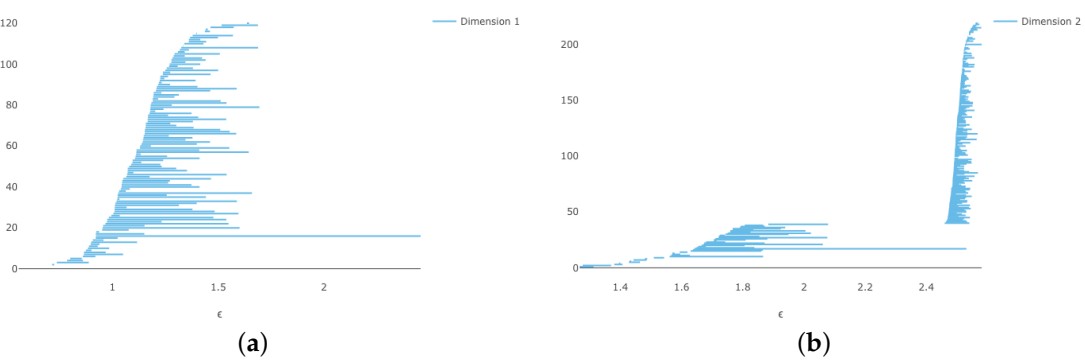

**Figure 9.** Vietoris–Rips persistence diagrams for 200 points on $\mathbb{R}P^3$ (**a**) $H_1$ persistence and (**b**) $H_2$ persistence.

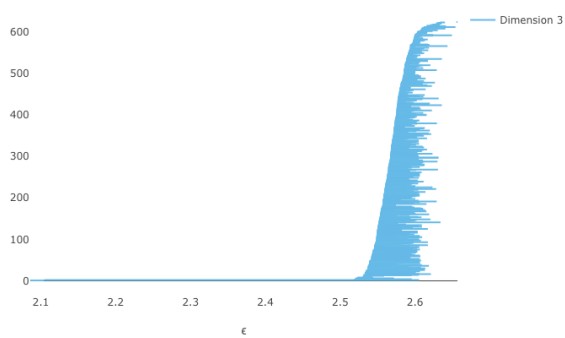

**Figure 10.** The $H_3$ barcode for 200 points on $\mathbb{R}P^3$.

### 3.4. $G_2(\mathbb{R}^4)$, Part I

We now consider the first Grassmannian that is not a projective space. Embed the 4-manifold $G_2(\mathbb{R}^4)$ as the space of symmetric idempotent $4 \times 4$ matrices of trace 2. As a first attempt, we take the naïve sampling approach of generating random pairs of orthonormal vectors to build a point cloud of such matrices. However, persistence calculations now become rather cumbersome. Table 1 shows some statistics on computation times for point clouds of various sizes on a MacBook Pro, 16 GB RAM.

**Table 1.** Computation times on a MacBook Pro, 16 GB RAM, of Vietoris–Rips persistence for various spaces. An X indicates that the software could not complete the calculation; a indicates the software was not used for the run. The ? indicates the number of simplices is unknown. Note the rapid explosion in the number of simplices.

| Space | # Points | Top Dim | # Simplices | Eirene | Ripser |
|---|---|---|---|---|---|
| $\mathbb{R}P^2 \subset \mathbb{R}^4$ | 100 | 2 | 206K | 0:00.53 | – |
| | 200 | 2 | 2.1M | 0:01 | – |
| $\mathbb{R}P^2 \subset \mathbb{R}^5$ | 100 | 2 | 436K | 0:02 | – |
| | 200 | 2 | 4.2M | 0:07 | – |
| $\mathbb{R}P^3 \subset \mathbb{R}^9$ | 100 | 3 | 7.6M | 0:09 | – |
| | 200 | 3 | 146M | 6:54 | – |
| $G_2(\mathbb{R}^4) \subset \mathbb{R}^{16}$ | 100 | 4 | 107M | 1:51 | 1:15 |
| | 150 | 4 | 792M | 1:04:45 | X |
| | 200 | 3 | 112M | 3:01 | 3:07 |
| | 200 | 4 | ? | X | X |

Eirene could compute homology for 200 points up to dimension 3 in about 3 min, producing a parameter value of $r = 0.95$ where the homology is correct in these dimensions. It seems that $H_4$ is the sticking point. The barcodes for 150 points are shown in Figures 11 and 12. At $r = 0.96$, the homology is correct up to dimension 3, but $H_4 = 0$ there.

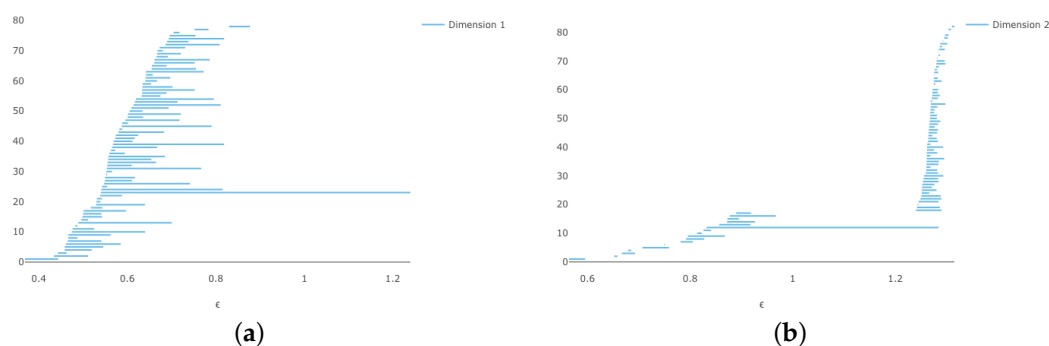

**Figure 11.** Vietoris–Rips persistence diagrams for 150 points on $G_2(\mathbb{R}^4)$ (**a**) $H_1$ persistence and (**b**) $H_2$ persistence.

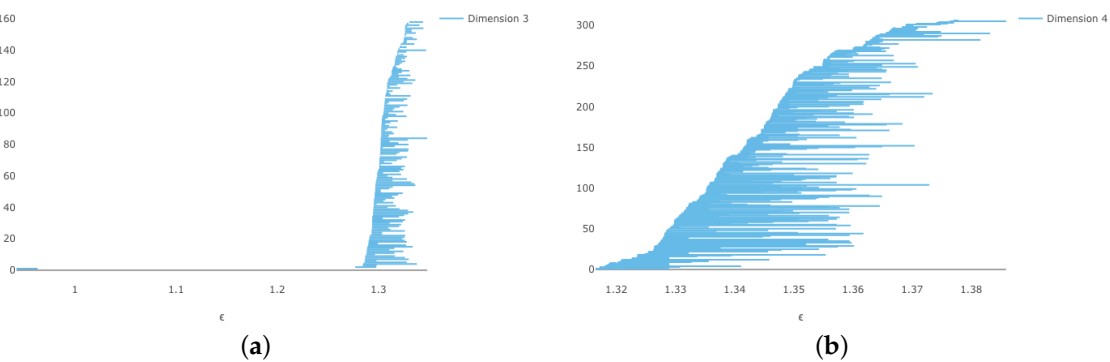

**Figure 12.** Vietoris–Rips persistence diagrams for 150 points on $G_2(\mathbb{R}^4)$ (**a**) $H_3$ persistence and (**b**) $H_4$ persistence.

In a quest for more memory, we received an offer from Mikael Vejdemo-Johannson to use his machine. It has 256 GB RAM. We began the 200 point Vietoris–Rips calculation in Eirene in the background and logged out. After 10 hours, it was still processing and was using 97% of the system memory. The next morning the process was complete; the output file (in JLD2 format) was 74 GB (!). Since Eirene uses PlotlyJS to render barcodes, they cannot be viewed remotely. Even if the file could be retrieved, it is unclear that our laptop could even open it, nor is there any guarantee that the barcodes are correct.

### 3.5. $G_2(\mathbb{R}^4)$, Part II

We then took a different approach. The Vietoris–Rips complex is nice because it is easy to compute, but it suffers from combinatorial explosion. We turned to witness complexes and made the associated computations using the Javaplex package [15] in MATLAB.

The initial attempt simply generated elements of $G_2(\mathbb{R}^4)$ by taking a pair of orthonormal vectors in $\mathbb{R}^4$ and using them to build a certain $4 \times 4$ matrix. For this experiment, we biased the sample in the following way. For a given number $M$ of points on $G_2(\mathbb{R}^4)$, we took 5% from the 1-cell, 15% from each of the 2-cells, 25% from the 3-cell, and 40% from the 4-cell. One could choose different proportions, of course.

This worked remarkably well. We generated 5000 points on $G_2(\mathbb{R}^4)$ and constructed the witness complex on 100 landmarks chosen using the max-min process. The barcodes for one such trial are shown in Figure 13. Note that we get the correct mod 2 and mod 3 homology for $r > 0.125$. This witness complex, which has 118,220 simplices, is therefore a good approximate triangulation of $G_2(\mathbb{R}^4)$. The point cloud and witness points are available as text files at https://github.com/niveknosdunk/grassmann. Note that the number of simplices in this witness complex is three orders of magnitude smaller than a Vietoris–Rips complex on the same number of points (cf. Table 1), so this construction is more effective all the way around.

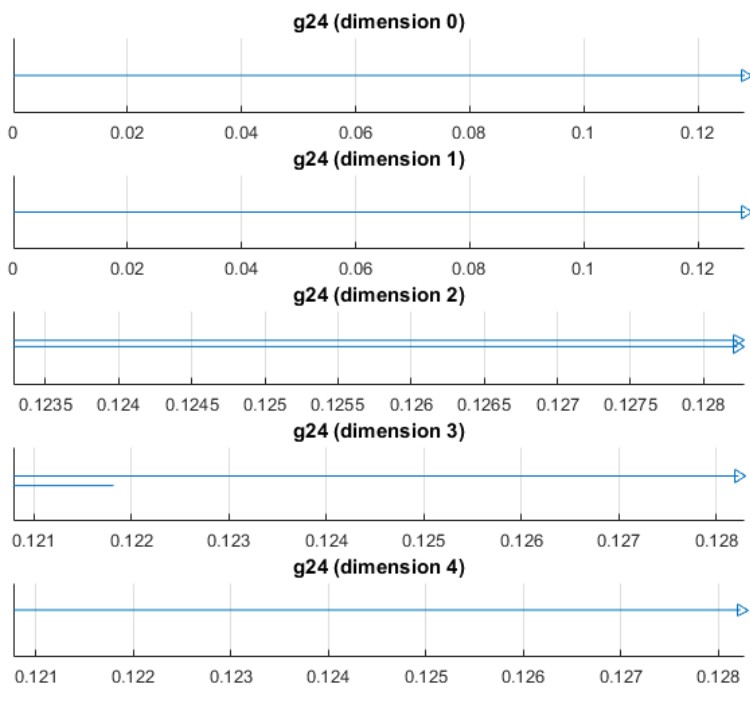

(**a**)

**Figure 13.** *Cont.*

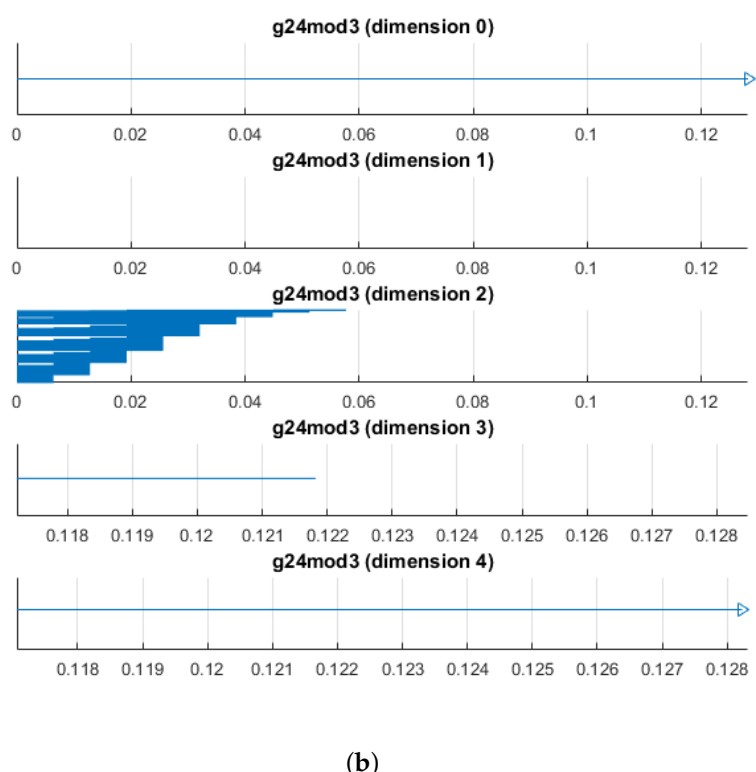

(**b**)

**Figure 13.** Barcodes for a witness complex on 100 points in a 5000-point sample on $G_2(\mathbb{R}^4)$ (**a**) $\mathbb{Z}/2$ coefficients and (**b**) $\mathbb{Z}/3$ coefficients.

## 4. Conclusions

In this paper, we demonstrated the utility of using Vietoris–Rips and witness complexes to obtain approximate triangulations of the Grassmann manifolds $G_k(\mathbb{R}^n)$. We were able to construct such spaces with relatively few vertices, but some questions remain for further study.

1. How small of a sample can we use to generate an approximate triangulation? For example, a result in [3] asserts that any triangulation of $G_2(\mathbb{R}^4)$ must have at least 14 vertices. We built an approximate triangulation using a witness complex on 100 landmarks. Surely, our algorithm will not work with only 14 points, but we plan to investigate how few we can get away with. A theorem of Niyogi–Smale–Weinberger [16] provides lower bounds on the number of points required to compute homology correctly with high probability, but these are certainly too high and can be improved in practice.

2. Can we push the computations further? The next Grassmannian to study is $G_2(\mathbb{R}^5)$. This is a nonorientable 6-manifold, and, using our procedure, we would embed it in $\mathbb{R}^{25}$. The machine used to compute the persistent homology of the witness complexes on $G_2(\mathbb{R}^4)$ in MATLAB ran out of memory on 100 landmarks in $G_2(\mathbb{R}^5)$. We therefore need either a bigger machine running MATLAB, or software that can handle witness complexes. The GUDHI package [17] is one option, but we have not attempted it yet.

3. The author expects to gain access to a new GPU based supercomputer at his institution in the next year. This may allow for similar computations on higher-dimensional $G_k(\mathbb{R}^n)$.

**Funding:** This research received no external funding.

**Acknowledgments:** This problem was suggested to me by Vidit Nanda; I thank him for the inspiration and helpful conversations. Henry Adams provided useful tips for Javaplex. I am also grateful to Mikael Vejdemo-Johansson for the use of his rather powerful computer. Marc Lange had a good suggestion about the proof of Proposition 2. Finally, thanks are due to a pair of anonymous referees for many useful comments.

**Conflicts of Interest:** The author declares no conflict of interest.

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
