# Peer review of "Approximate Triangulations of Grassmann Manifolds"

_algorithms, doi:10.3390/a13070172_

Round 1

Reviewer 1 Report

In this article, the author demonstrates how to compute approximate triangulations of Grassmann manifolds. The idea presented is to first sample the Grassmann manifold using its Schubert decomposition, then compute the associated Vietoris-Rips / witness complex of the point cloud, and stop at a radius for which the homology of the Grassmann manifold and the one obtained from the bars in the persistence barcode agree. A bunch of examples is provided, either using explicit embeddings of projective spaces, or diffeomorphisms on the space of idempotent symmetric matrices of trace k.

Overall, this article is very easy to follow and demonstrates interesting experiments from a nice idea. However, I am concerned about a few things:

1. Topological spaces can have same mod 2 homology groups while being extremely different, like the torus and the Klein bottle (in dimension 0 and 1). So it is not clear to me whether only checking mod 2 homology is enough (I guess knowing that the vertices are all part of the Grassmann manifold is an additional guarantee somehow). Why not checking homology with other coefficients? This might be more convincing (I think Gudhi can do it).

2. The paper is well written but it lacks a bit of motivation (from my naive point of view): why is it important to triangulate Grassamann manifolds? Is it a pure math question or does it have practical applications? If so, would it be possible to provide a couple examples where Grassmann manifolds are actually used?

3. The comparison with Ripser is nice. I think it could be interesting to extend the table by comparing against a recent paper that efficiently computes persistence of flag complexes: https://drops.dagstuhl.de/opus/volltexte/2019/10459/

I like the work but I find it difficult to understand how good is the actual triangulation and how useful is the problem.

Author Response

I thank the referee for a careful reading of the manuscript and for useful comments. I have addressed the concerns as follows.

  1. The suggestion to compute the mod 3 homology is a good one. I added the calculation of the integral and mod 3 homology of G_2(R^4) in Section 2 and included the mod 3 barcodes for this space. I did not add the mod 3 barcodes for the projective spaces as to not clutter the paper further with more figures. The Grassmannian is the more interesting example.
  2. This is more of a pure math question being attacked using a tool from applied topology. I added a discussion about some open questions about triangulations of Grassmannians (line 21) and a paragraph about the overall project (line 42). 
  3. I expanded Table 1 to include all the Vietoris-Rips examples and added the number of simplices in each complex. This indicates why the VR calculations fail so quickly--many of these complexes have hundreds of millions (!) of simplices. 

Reviewer 2 Report

Computational topology generally makes use of theoretical techniques for solving application problems. This paper goes the other way round: It adapts a tool - conceived for applications - to a theoretical problem, the one of economic triangulations of Grassmanian manifolds.

Let me clarify this point. Generally, Vietoris-Rips complexes and witness complexes (among others) are built upon a cloud of sampling points of an (unknown) object X, depending on a positive parameter alpha. The basic idea of persistence is to take into account the homology of the complexes by varying alpha. The homology cycles which "persist" for longer alpha intervals are thought to represent the true, wished for homology of X.

Here the homology of the objects of interest is known a priori: the theory of Grassmanian manifolds is rich enough. The author wants to generate economic (approximate) triangulations of these manifolds. So he makes use of a smart embedding of the Grassmanian manifolds into Euclidean spaces, produces a point cloud sampling them and lets alpha grow. Then he choses an alpha such that the corresponding complex has the right homology. Vietoris-Rips turns out to be computationally too expensive, so the author switches to the more economic witness complex.

All in all, the paper is very interesting and original; the idea is smart and well described; the paper also succeeds in being self-contained. I definitely propose to accept it for publication. Just two small remarks:

l.57-59
I think these lines would be better placed immediately after lemma 4 (l.51).

l.70 and caption of Fig. 1
tetrahedron --> tetrahedron boundary

Author Response

I thank the referee for a careful reading of the manuscript. I made the two suggested changes, moving the table of Schubert symbols before Lemma 4, and adding "boundary of the" to "tetrahedron" in the two places indicated.

Round 2

Reviewer 1 Report

The new version is fine for publication.